# When face masks signal social identity: Explaining the deep face-mask divide during the COVID-19 pandemic

**Nattavudh Powdthavee**[1]◉*, **Yohanes E. Riyanto**[2]◉, **Erwin C. L. Wong**[2]◉, **Jonathan X. W. Yeo**[2]◉, **Qi Yu Chan**[2]◉

**1** Warwick Business School, Coventry, United Kingdom, **2** Nanyang Technological University, Singapore, Singapore

◉ These authors contributed equally to this work.
* Nattavudh.powdthavee@wbs.ac.uk

**Data Availability Statement:** The project was pre-registered on 2nd December 2020 on Open Science Framework (OSF: https://osf.io/qmyg5/). The study's data, STATA .do files, and README.txt

## Abstract

With the COVID-19 pandemic still raging and the vaccination program still rolling out, there continues to be an immediate need for public health officials to better understand the mechanisms behind the deep and perpetual divide over face masks in America. Using a random sample of Americans ($N = 615$), following a pre-registered experimental design and analysis plan, we first demonstrated that mask wearers were not innately more cooperative as individuals than non-mask wearers in the Prisoners' Dilemma (PD) game when information about their own and the other person's mask usage was not salient. However, we found strong evidence of in-group favouritism among both mask and non-mask wearers when information about the other partner's mask usage was known. Non-mask wearers were 23 percentage points less likely to cooperate than mask wearers when facing a mask-wearing partner, and 26 percentage points more likely to cooperate than mask wearers when facing a non-mask-wearing partner. Our analysis suggests social identity effects as the primary reason behind people's decision whether to wear face masks during the pandemic.

## Introduction

On 11 March 2020, the World Health Organization (WHO) declared COVID-19 a global pandemic, which signifies that the virus had spread worldwide. In response to the WHO's announcement, many governments worldwide started recommending or mandating their citizens to work from home, maintain social distance, wash hands regularly, and wear face masks or face-covering in public places [1]. Yet, despite the latest scientific evidence concluding that mass masking is one of the most effective public health strategies to curtail the COVID-19 transmission in the community [2–5], a significant number of people in the U.S. have, regrettably, continued to reject explicit rules about wearing face masks [6, 7]. For example, YouGov reported in January 2021 that around 25% of Americans continued to refuse to wear face masks in public places [8].

have been deposited on the same project page on the OSF website.

**Funding:** Nattavudh Powdthavee received funding from Warwick University's and Yohanes E. Riyanto and Jonathan W.X. Yeo received funding from Nanyang Technological University's personal research budgets for this work.

**Competing interests:** The authors have declared that no competing interests exist.

Why do many individuals continue to disregard public health regulations designed to protect themselves and others in the community? One hypothesis is that people with different moral concerns react differently to the recommended guidelines. In an online study of Americans, Chan [9] shows that people who have strong preferences for caring for others and equality for all are more likely to adhere to the COVID-19 recommended guidelines, including wearing face masks in public places. Another hypothesis is that mask-wearing in America became so politicized early on in the pandemic due to the government officials' inconsistent face mask policies, which caused a divide in people's attitudes towards face masks along the political line [10, 11]. For instance, Yeung et al. [12] show using Twitter data that there was a sharp and persistent increase in polarization between Democrats' and Republicans' sentiment towards mask-wearing that took place on 3 April 2020 when the Centers for Disease Control and Prevention (CDC) reversed its earlier stance in favour of personal face mask usage. Rothgerber et al. [13] demonstrate that political conservatism is one of the most significant predictors of non-compliance with the face mask rules in the U.S. Furthermore, a nationally representative study conducted in April-May 2020 shows that Democrats were significantly more likely to perceive more risk associated with COVID-19 and, therefore, were more willing to engage in protective behaviours, including wearing face masks, than Republicans [14].

Social perception towards people who wear face masks may also determine the face mask usage rate in society. Does wearing a face mask signal to other people how much we care about protecting ourselves and the community we live in from COVID-19? Or does it signal our political identity [15]? By understanding what social cues either wearing or not wearing a face mask sends to other people, policymakers can potentially design more effective public messages to increase compliance among non-mask wearers. For example, assume that people generally perceive mask wearers as individuals who have strong preferences for caring for others and non-mask wearers as selfish individuals. Assume also that people cooperate more with those who are cooperative, then providing evidence in support of these two assumptions to non-mask wearers might give them additional incentives to start wearing face masks. By contrast, if wearing face masks signals primarily one's political identity to others, then disloyalty aversion [16, 17] and in-group altruism [18] imply that it would be psychologically challenging for non-mask wearers to start cooperating and complying with the official guidance as doing so would signal that they are not as loyal to their identity as they believe they are. In this case, public messages that emphasize how wearing face masks show caring are unlikely to change non-mask wearers' behaviours. Here, a better appeal might be one that depoliticizes and, in effect, destigmatizes mask-wearing among the non-mask wearers altogether.

In this high-powered, pre-registered study (OSF: https://osf.io/wq47e), we contribute to the growing literature that uses social dilemma games to model people's behaviours in a pandemic situation [19–21] that includes, for example, individuals' strategic decision to take up a vaccine [22], and policy makers' decision of whether to continue shutting down the economy to curtail the spread of COVID-19 [23]. We do so by experimentally investigating the social perception of mask and non-mask wearers during the COVID-19 pandemic in America. Instead of simply asking what mask and non-mask wearers think of each other, we look at how information about others' mask-wearing behaviours influences cooperation when real money is at stake. Our experimental design, which we describe in detail in the next section, uses a series of simultaneous and sequential Prisoner's Dilemma (PD) games as a setting to test the extent to which people perceive others' mask-wearing behaviour as a signal of their willingness to cooperate or their political identity. We believe our results could help inform policy makers of the motivations behind individuals' decision to shun face masks and, subsequently, help them design the most appropriate public health strategy to encourage non-mask wearers to start wearing face masks to protect themselves and others in the community.

## Materials and methods

### Participants

Subjects from the US were recruited via Prolific, to participate in a study about social preferences. Given the politicization of face mask usage, we chose to recruit an even distribution of subjects from different political affiliations. In total, 615 participants completed the study between 9th and 14th December 2020.

Of the 615 participants, 59 (10%) were non-mask wearers, 318 (52%) were males, 214 (35%) were Democrats, and 215 (35%) were Republicans. See S1 Table for more summary statistics. The relatively small number of non-mask wearers in our sample (N = 59) is fairly representative of the ~80% face mask usage rate in America recorded in December 2020 [24], which was the same period as when we ran our study. Had we run our study in early 2020, we would have been able to recruit a significantly higher number of non-mask wearers for our analysis.

### Experimental details

Our experiment consisted of two main stages where participants played a series of incentivized Prisoner's Dilemma (PD) games—with and without—knowing their partners' mask usage, see Fig 1, Panel 2. In every PD game, participants were shown the PD payoff table in Fig 1, Panel 1 and had to choose whether to cooperate (A) or to defect (B) against an opponent. We selected the PD game as it represents the standard analytical tool for studying cooperative behaviours in a social dilemma setting [25, 26]. As defection is a dominant strategy, theory predicts that mutual defection would prevail. Willingness to refrain from defecting by embracing cooperation requires a leap-of-faith for participants to trust their opponent to reciprocate.

In the first stage, participants played a series of three PDs. In the first PD, participants and their partners simultaneously chose their actions. However, the second and third PDs were sequential—that is, subjects chose their actions conditional on their partners choosing either A

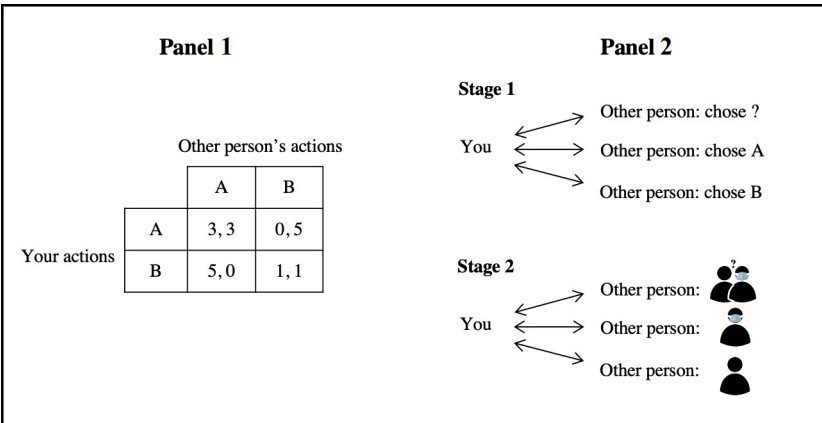

**Fig 1. Experimental setting for the Prisoner Dilemma (PD) games.** Panel 1 depicts the actions and associated payoffs for players in the incentivized PD game. Action A represents the choice to cooperate, while action B represents the choice to defect. The player's payoff lies leftwards of the comma in each cell, while the other person's payoff lies rightwards. Participants played the PDs either simultaneously or sequentially. Panel 2 illustrates Stages 1 and 2 of the PD games. In Stage 1, participants played a series of three PDs without mask wearing being salient. In the first PD game, participants choose their actions simultaneously, and in the second and the third PD game, they choose their actions conditional on their partners having chosen A or B, respectively. In Stage 2, participants played a series of three PDs after mask-wearing was made salient to them. In these three PD games, participants choose their actions simultaneously against i) an anonymous partner, ii) a mask-wearer partner, and iii) a non-mask-wearer partner. The three PD games were randomly ordered.

or B, respectively. Here, no one had any information about whether partners were mask wearers; neither was mask wearing salient to them.

After the first stage ended and before the second stage began, participants were asked about their mask wearing decisions given a fixed hypothetical scenario involving COVID-19. Each subject's answer to the question posed in the scenario was used to classify them as a mask wearer (MW) or a non-mask wearer (NMW).

In the second stage, participants played another series of three PDs in a random order against i) an anonymous partner, ii) a MW partner, and iii) a NMW partner. In all three PDs, actions were chosen simultaneously. In contrast to the first stage, participants playing the PDs in the second stage either had direct information about whether partners were mask wearers (i and ii); or had mask wearing made salient to them (iii).

At the end of the experiment, we also elicited participants' altruism towards mask wearers and non-mask wearers, and beliefs about mask wearers' and non-mask wearers' willingness to cooperate in an incentivized manner. We then asked participants to report standard socio-demographic questions, and their general attitudes towards mask-wearing in general. For more details about the experiment, refer to the Experimental Details file that we uploaded to the Open Science Framework's project website: https://osf.io/xkt5c/.

## General experimental procedure

This research was reviewed and given full approval by the University of Warwick Research Governance and Ethics Committee and also the Nanyang Technological University (NTU) Institutional Review Board (IRB). All authors confirm that the experiment was performed in accordance with the University of Warwick Research Governance and Ethics Committee's and the NTU IRB's guidelines and regulations.

The experiment was pre-registered on Open Science Framework: https://osf.io/wq47e. It was programmed in oTree [27] and hosted online via a web application using Heroku, a cloud application platform (https://www.heroku.com/). The web application was deployed to the American public via the Prolific website (www.prolific.co).

After obtaining participants' consent to take part in the study, participants were directed to the instructions of the experiment, after which they began the PD games. Participants were given detailed instructions on all tasks prior to starting each section. We avoided framing whenever possible by using neutral terms such as 'participants', 'task', and 'other participant you are paired with' instead of words like 'players', 'game', and 'partner'.

The sessions, which comprised a total of 615 participants, were run in four waves from 9 December 2020 to 14 December 2020 between 11am and 2pm (GMT-5) to account for the different time-zones in the U.S. The median completion time of the experiment was approximately 18 minutes. Participants were told that they would be paid a show up fee and a monetary bonus based on the Experimental Currency Unit (ECU) they had accumulated throughout the study. The first 182 participants in the earlier sessions had a show-up fee of $3 and an exchange rate of 10 ECU per dollar. The other 433 participants in the later sessions had a show-up fee of $2.35 and an exchange rate of 5.5 ECU per dollar. The median payment of all participants was $4.42, while the median payment of the 433 participants was $4.20.

## Hypotheses

We set out to test the following hypotheses:

*H1: Mask wearers are more willing to cooperate than non-mask wearers in the PD games when information about each other's mask usage is not revealed.*

Based on the findings that people who have strong preferences for caring for others and equality for all are more likely to adhere to the COVID-19 guidelines [9], H1 predicts that face mask wearers are more likely to cooperate in the PD games than non-mask wearers when the information about the other player's mask usage is not made publicly available.

*H2: Revealed information about other player's face mask usage signals their willingness to cooperate in the PD games. Consequently, mask wearers are more likely to cooperate with other mask wearers but defect against non-mask wearers, while non-mask wearers are more likely to defect against both mask and non-mask wearers.*

Assuming that others perceive mask wearers to be generally more cooperative than non-mask wearers [9] and that people are generally conditional cooperators [28], H2 predicts that participants are likely to use the revealed information about other people's mask usage as a signal of willingness to cooperate in the PD games. Given that mask wearers expect other mask wearers to cooperate, H2 implies that mask wearers will likely cooperate when playing against each other and defect when playing against other non-mask wearers who are expected to defect. On the other hand, H2 predicts that non-mask wearers will likely defect when playing against both mask and non-mask wearers. This is because non-mask wearers will expect (i) mask wearers to defect when playing against non-mask wearers and (ii) other non-mask wearers to defect in general. Hence, if H2 is true, then the best response for non-mask wearers will be to defect when playing against either mask or non-mask wearers.

*H3: Revealed information about other player's face mask usage signals their social identity. Consequently, mask wearers are more likely to cooperate with other mask wearers but defect against non-mask wearers, while non-mask wearers are more likely to defect against mask wearers but cooperate with other non-mask wearers.*

According to social identity theory [29], in which an individual's self-concept is derived from membership in a relevant social group, H3 predicts that because of the effect of social identity on social preferences and beliefs [30], mask wearers will be more likely to cooperate when playing against another mask wearer (in-group) and defect when playing against a non-mask wearer (out-group). On the other hand, non-mask wearers will be more likely to cooperate when playing against another non-mask wearer (in-group) and defect when playing against a mask wearer (out-group).

As shown in Table 1, the only difference between H2's and H3's predictions can be found in the bottom right-hand corner. Here, evidence of non-mask wearers choosing to defect against another non-mask wearer would support the hypothesis that face masks signal cooperation and compliance to others. On the other hand, evidence of non-mask wearers choosing to

**Table 1. Predictions of mask and non-mask wearers' cooperative behaviours when information about the other player's mask usage is revealed.**

| Individual $i$'s Cooperative behaviour | Mask usage | Playing against a mask wearer | Playing against a non-mask wearer |
|---|---|---|---|
| | *Mask wearer* | H2: + | H2: - |
| | | H3: + | H3: - |
| | *Non-mask wearer* | H2: - | H2: - |
| | | H3: - | H3: + |

H2 assumes that face masks signal cooperation, while H3 assumes that face masks signal social identity. The sign '+' indicates that the average cooperation level is higher and '-' is lower than the baseline, where information about the other person's face mask usage is not revealed.

cooperate more with another non-mask wearer would support the hypothesis that face masks signal social identity to others.

Note that there is a possibility that face masks signal **both** cooperation and social identity, in which case the opposing effects may cancel each other out, and non-mask wearers' cooperation level will be similar regardless of whether they are playing against another non-mask wearer or a mask wearer. However, we should continue to observe mask wearers cooperating more when playing against another mask wearer where both H2 and H3 are true.

## Results

Previous evidence suggests that people who have strong pro-social preferences are more likely to wear face masks in public places [9]. Yet, there is little evidence in the first stage of our experiment—before we elicited their face mask usage—that mask wearers were more prosocial than non-mask wearers when they were playing the PD games against an anonymous partner. As can be seen in Fig 2, mask wearers and non-mask wearers did not exhibit statistically significant differences in cooperation with anonymous partners.

Fig 2, Panel 1 shows that when playing the PD game simultaneously, non-mask users cooperated 69% of the time compared to 71% of the time for mask wearers (Fisher's exact test, $p = 0.881$). Fig 2, Panels 2 and 3 also show that when playing the PD game sequentially, both mask and non-mask wearers cooperated more following a partner's decision to cooperate (66% and 75% respectively) than when following a partner's decision not to cooperate (22% for both). In both cases, the differences across mask and non-mask wearers were not statistically significant (Fisher's exact test, $p = 0.194$ for the former and $p = 1.000$ for the latter). Hence, there is little evidence to support H1 that mask wearers are intrinsically more cooperative than non-mask wearers when information about their own and the other person's mask usage is not salient.

We then proceeded by asking participants about their face-mask usage in the second stage of the experiment. Following this elicitation, participants played three simultaneous PD games against i) an anonymous partner, ii) mask-wearing partner, and iii) non-mask wearing partner in random order. Next, we compare the level of cooperation that someone shows towards an anonymous partner before and after eliciting information about face mask usage; see Panel 1 of Figs 2 and 3, respectively. Fig 3, Panel 1 shows 69% of mask wearers and 66% of non-mask wearers cooperated when playing against an anonymous partner. Differences in cooperation before and after asking about face mask usage were not significant for both mask and non-mask wearers (Wilcoxon signed-rank test, $p = 0.307$, and $p = 0.804$ for mask and non-mask

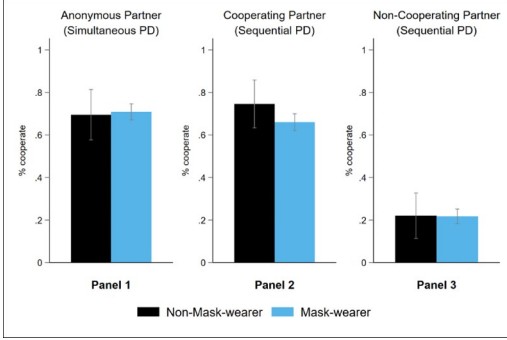

**Fig 2. Average cooperation levels in PD games before mask usage elicitation.** 4-standard-error bars (2 above, 2 below) to represent 95% confidence intervals.

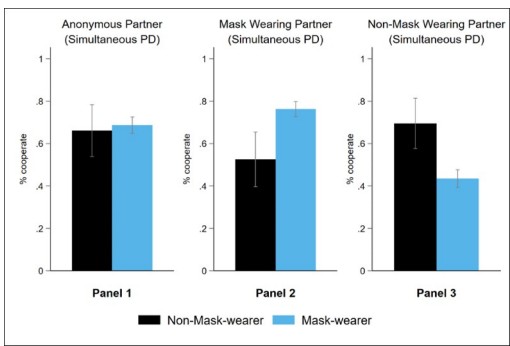

**Fig 3. Average cooperation levels in PD games after mask usage elicitation.** 4-standard-error bars (2 above, 2 below) to represent 95% confidence intervals.

wearers). Neither were the differences between the mask and non-mask wearers in the second stage (Fisher's exact test, p = 0.662).

The elicitation of participants' mask usage after the first stage, which would have made their own mask-wearing behaviour salient, did not affect cooperation in the PD game. This result suggests that participants do not generally hold stereotypical views about how mask or non-mask wearers should behave in a social dilemma situation, which is inconsistent with H2. While mask and non-mask wearers were equally likely to cooperate when information on mask-usage was not revealed, we found significant between-group differences in cooperation when partner's mask-usage was revealed to the participants; see Panels 2 and 3 in Fig 3. When playing against another mask wearer, 76% of mask wearers, as opposed to 53% of non-mask wearers, cooperated. In contrast, when playing against another non-mask wearer, 44% of mask wearers, as opposed to 69% of non-mask wearers, cooperated in the PD game.

These aggregate numbers are more consistent with H3 than H2 in that participants generally viewed face masks as a signal of one's social identity, which could have generated the "us vs. them" feelings between in-group and out-group members. Here, we have raw data evidence to suggest that mask wearers treated other mask wearers as in-group members and non-mask wearers as out-group members, and vice versa for how non-mask wearers treated other non-mask wearers and mask wearers in the PD game.

Fig 3's results continue to be robust even after controlling for gender, age, ethnicity, the political party supported, education, household income, the exchange rate, and the order of the PD games in a pooled probit regression, see S2 Table. Holding other things constant, we found that non-mask wearers were 2 percentage points less likely to cooperate compared to mask wearers when facing an anonymous partner, although the difference was statistically insignificantly different from zero ($p = 0.724$). By contrast, non-mask wearers were 23 percentage points ($p = 0.001$) less likely to cooperate than mask wearers when facing a mask-wearing partner, and 26 percentage points ($p < 0.001$) more likely to cooperate than mask wearers when facing a non-mask wearing partner.

Not only did we find substantial differences in mask and non-mask wearers' cooperation rates within mask and non-mask wearing partner conditions, but we also uncovered evidence of a significant variation in mask wearer and non-mask wearers' cooperation rates across different conditions. More specifically, compared to facing an anonymous partner, we found mask wearers' cooperation rates were 7 percentage points ($p < 0.001$) higher when facing a mask-wearing partner and 25 percentage points ($p < 0.001$) lower when facing a non-mask-wearing partner. The difference of 32 percentage points is statistically significant in a $t$-test, $p < 0.001$. In contrast, we found that non-mask wearers' cooperation rates were 14 percentage

points ($p$ = 0.063) lower when facing a mask-wearing partner and 3 percentage points ($p$ = 0.634) higher when facing a non-mask-wearing partner, compared to facing an anonymous partner. The difference of 17 percentage points is significant in a $t$-test, $p$ = 0.033. Taken together, we have enough evidence from the raw data and multivariate analysis to support the hypothesis that face masks strongly signal social identity and that individuals are more likely to cooperate with in-group members and defect against out-group members.

Incentivized questions at the end of the experiment lend further support to H3, which is based on the hypothesis that people of the same identity groups are likely to favour members of one's in-group over out-group members. More specifically, we found that mask wearers (non-mask wearers) were 1) more altruistic towards mask wearers (non-mask wearers), 2) more likely to believe that other mask wearers (non-mask wearers) would cooperate with them, and 3) more likely to believe that other mask wearers (non-mask wearers) expect more cooperation from them in the PD game, see S1 Fig. We also show that these elicited expectations about the mask and non-mask wearers' willingness to cooperate and their altruism towards them are strongly correlated with participants' cooperation; see S3 Table.

To what extent can we attribute Fig 3's evidence of social identity effects from mask usage to the evidence of political polarization of face masks in America [10–14]? Looking at the raw data, we can see evidence of significant disparity in mask usage by political affiliation in our sample; 98% of Democrats identified themselves as mask wearers compared to 88% of Independents and 84% of Republicans. These differences are not only sizable but also statistically significant; we can reject the null hypothesis that the average mask usage is the same across all three political affiliations; Fisher's exact tests between Democrats-Independents and Democrats-Republicans produce $p$-values<0.001. We also uncovered evidence that participants supporting different political parties had very different world views. In the post-experiment survey, we found Democrats were more likely to report greater concern for the Covid-19 situation, and that mask-wearing was a "public health responsibility, the "right thing to do", and an effective anti-COVID19 measure compared to Independents or Republicans. Interestingly, Democrats had more favourable opinions of mask wearers' morals over non-mask wearers' morals than Independents or Republicans; see S4–S6 Tables for more details. These aggregate numbers lead us to form an additional hypothesis about the sources of social identity effects as follows:

*H4: Social identity derived from face mask usage can be primarily attributed to one's political affiliation.*

H4 rests on the assumption that people are aware that there is a higher proportion of mask wearers who are Democrats than non-Democrats and, through representativeness heuristic [31], perceive that wearing a mask is more representative among Democrats and not wearing a mask is more representative among non-Democrats, i.e., face mask usage is a signal of political identity.

Fig 4 provides raw data analysis of H4 by examining how the bias towards mask wearers over non-mask wearers differs across the political spectrum. It shows that bias towards mask wearers over non-mask wearers was highest among Democrats, followed by Independents, and then Republicans; see Fig 4, Panel 1. Likewise, the bias towards mask wearers over non-mask wearers decreases with how conservative participants think of themselves, see Fig 4, Panel 2.

To formally test H4, we included interaction terms between an individual's political affiliation and partner's mask usage in the fully-specified probit regression where cooperation is the outcome variable, see S7 Table, Panel A. When facing a mask-wearing partner, Independents' and Republicans' cooperation rates were 5 percentage points ($p$ = 0.334) and 10 percentage

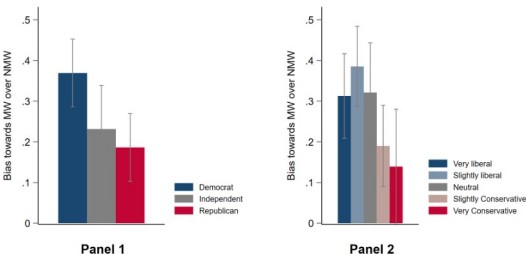

**Fig 4. Bias towards Mask Wearers (MW) over Non-Mask Wearers (NMW) by political affiliation.** Bias is calculated as the difference in average cooperation levels toward mask wearers (MW) and non-mask wearers (NMW) in the simultaneous PD games. 4-standard-error bars (2 above, 2 below) to represent 95% confidence intervals.

points ($p$ = 0.015) lower than Democrats'. When facing a non-mask-wearing partner, Independents' and Republicans' cooperation rates were instead both 8 percentage points ($p$ = 0.151 and p = 0.109 respectively) higher than Democrats'. The lower cooperation rates towards mask wearers compared to non-mask wearers of 13 percentage points for Independents and 18 percentage points for Republicans are statistically significant in $t$-tests, $p$ = 0.058, and $p$ = 0.002. Similar results also hold for the interaction between a mask-wearing partner and political conservativeness, see S7 Table, Panel B.

Moreover, we also found a similar interaction effect of political affiliation and partner's mask usage on participant's social preferences and beliefs. Compared to Republicans or Independents, Democrats were more altruistic (less altruistic) towards other mask wearers (non-mask wearers). Furthermore, in almost all comparisons to Republicans or Independents, Democrats believed that other mask wearers (non-mask wearers) were more likely (less likely) to cooperate with them and to expect more cooperation (less cooperation) from them than non-mask wearers (mask wearers) in the PD game; see S8 Table. Regressions that examine the interaction between political conservativeness also yield qualitatively similar results; see S9 Table.

What happens to the interaction effects between own and partner's face mask usage on cooperation once we take into account the interplay (interaction) between political affiliation and partner's mask usage? If H4 is true, then including this interplay should drive the interaction effects between own and partner's face mask usage on cooperation towards zero. While including the new interaction terms reduced the size of the interaction effects between own and partner's mask usage on cooperation, the effects continued to be quantitatively important as well as statistically significant; see S10 Table. For example, non-mask wearers were still 21 percentage points ($p$ = 0.002) less likely to cooperate than mask wearers when facing a mask-wearing partner, and 25 percentage points ($p$<0.001) more likely to cooperate than mask wearers when facing a non-mask wearing partner even after taking into account the interaction terms between political affiliation and partner's mask usage in the regression.

In summary, while there is evidence that political identity drives some of our earlier results of social identity effects associated with face mask usage, we cannot attribute the entire effects to the hypothesis that face masks are a pure signal of political identity. There is still a residual effect that cannot be accounted for by the political identity, which implies that face masks also carry an identity, one that is distinct from political identity, that is capable of imparting a sense of unity among mask wearers.

This subtle notion of identity could be explained using the insights from the literature on minimal groups [32] where the mere identification of individuals to groups—even if on an arbitrary basis—suffices to induce "us-*vs.*-them" effects. It is plausible that strong perceived political identity may have served to amplify such minimal group effects in our experiment.

This might be related to the considerably larger bias towards in-group members over out-group members in our experiment of 31 percentage points, compared to the typical social identity effects of around 14–19 percentage points found in previous studies [18, 33]. An alternative explanation of the residual effect might be that mask-wearers (non-mask wearers) have a strong distaste against exhibiting anti-social (pro-social) behaviours. Consequently, mask-wearers (non-mask wearers) might want to engage in pro-social (anti-social) punishment against non-mask wearers (mask wearers).

## Discussions

In this paper, we conducted a pre-registered, online incentivized lab experiment using a high-powered sample of Americans to test whether people generally use others' face mask usage as a signal of social identity instead of innate willingness to cooperate during the COVID-19 pandemic. In contrast to previous research that used surveys to demonstrate that mask wearers have strong preferences for caring for others and equality for all [9], we found little evidence that mask wearers behaved more cooperatively than non-mask wearers in the PD game compared to non-mask wearers when information about their own and the other person's mask usage is not salient. However, more consistent with social identity theory, we found strong evidence of in-group versus out-group bias based on mask usage during the pandemic. Non-mask wearers were 23 percentage points ($p$ = 0.001) less likely to cooperate than mask wearers when facing a mask-wearing partner, and 26 percentage points ($p$<0.001) more likely to cooperate than mask wearers when facing a non-mask wearing partner. These findings are surprising, considering that there is little evidence that mask wearers were generally more cooperative than non-mask wearers in the scenario where the information about mask usage was not known to the participants.

Our results are notably different from recent studies that found zero social identity effects associated with COVID-19 vaccination. Kohn et al. [34] demonstrated that vaccinators and non-vaccinators generally treat vaccinators better in prosocial activities and, in a follow-up study by Weisel [35], that there is little evidence of the politicization of vaccination in people's prosocial behaviours even when, like face masks, there are more Democrats than Republicans who are pro-vaccine. One possible explanation for this is that, unlike face masks, vaccination and vaccination intentions are not readily visible to both in-group and out-group members. Hence, despite evidence of political partisanship based on vaccination against COVID-19 shown in Weisel [35], without visibility of one's vaccination status to others, it would be less likely that vaccination is going to be affected by politicization when compared to wearing face masks in America [10–14].

Moreover, not only have we demonstrated that face masks signal strong social identity, we have also uncovered evidence of an in-group bias based on face masks that is completely orthogonal to one's political identity and remains unexplained in our regression model. Despite the politicization of face masks being the likely root cause of the social identity effects, our experimental evidence seems to suggest that people may have evolved over time to assign face mask usage as a minimal condition required for favouring in-group members and discriminating against out-group members.

Our results, which provide new insights into the extent and the mechanisms behind the deep divide over face masks in America, have important public health implications. With the more infectious strains, e.g., the UK (B.1.1.7) and South African (B.1.351) variants, taking over and vaccination programs still rolling in America, public messages designed to curb the transmission rate by increasing awareness about face mask effectiveness in protecting themselves and others in the community from COVID-19 [36] are unlikely to change non-mask wearers'

world views and behaviours towards mask usage as doing so would signal disloyalty to their held political identity. A better public health strategy might focus less on the details of the messages and more on the 'messenger' or the information source. Studies in behavioural economics have shown how messengers who are authority figures, share similar characteristics with and are likable to the target individuals, tend to be more successful in getting their messages across and, in turn, change individuals' choices and behaviours [37, 38]. Given that part of the social-identity effects is explained by political identity, non-mask wearers might be more willing to listen to a message about face mask's effectiveness from an authoritative figure in the Republican party or non-political figures who share similar characteristics or are generally well-liked by non-mask wearers. This could include, for example, family doctors of non-mask wearers and mask-wearing friends who share the same political affiliation as non-mask wearers. Future research could explore what type of messenger works best at reducing the social identity effects of face masks and, in-so-doing increase the mask usage rate in the United States and elsewhere around the world.

Finally, our findings, when viewed in conjunction with Korn et al. [34] and Wiesel [35], suggest that political partisanship based on health measures are more likely to lead to actual polarization in the take-up rate when the health measures in question are visible and salient to the individual and others in the community. Given the recent political discussions on vaccine passports [39, 40] and getting those who have been vaccinated against COVID-19 to wear a sticker visible to others [41], our results suggest that such efforts might lead to the unintended politicization of vaccinations, which would inevitably undermine the large-scale vaccination efforts to stop the spread of COVID-19.

Like all studies in social sciences, our study is not without limitations. One concern is the external validity of our findings. While it has been shown that people can easily identify PD games and play according to the game theory in the lab [42], it is possible that the same individuals may behave differently when facing a similar social dilemma in the real-world. It also remains to be seen whether our results can be generalised to other types of health measures such as vaccine passports and social distancing—scenarios where the stakes are large and interactions are repeated across countries and stages of the pandemic. Nonetheless, we have no reason to believe that the results depend on other characteristics of the subjects, materials, or context that are not already accounted for in the current study.

## Supporting information

**S1 Fig. Expectations and altruism towards partners by mask usage.** 4-standard-error bars (2 above, 2 below) to represent 95% confidence intervals.
(DOCX)

**S1 Table. Summary statistics for key demographic variables.** SD in parentheses.
(DOCX)

**S2 Table. Marginal effects within and across conditions.** * 0.10 ** 0.05 *** 0.01. Standard errors in parentheses, clustered at individual level. Marginal effects from a Pooled Probit Regression using data on cooperation towards mask wearers, non-mask wearers and anonymous partners. For Panel A, marginal effects are relative to that of having an anonymous partner. For Panel B, marginal effects are relative to that of being a mask wearer. Includes controls for gender, age, ethnicity, the political party supported, education, household income, the exchange rate, and the order of the PD games.
(DOCX)

**S3 Table. Mediators of cooperation.** * 0.10 ** 0.05 *** 0.01. Errors clustered at individual level. Marginal effects from a Pooled Probit Regression using data on cooperation towards mask wearers and non-mask wearers. Includes controls for gender, age, ethnicity, the political party supported, education, household income, the exchange rate, and the order of the PD games.
(DOCX)

**S4 Table. Opinions on Covid-19 by political affiliation.** * 0.10 ** 0.05 *** 0.01. OLS regressions with controls for own mask usage, gender, age, ethnicity, education, household income, the exchange rate, and the order of the PD games. Baseline group is Democrats. See S5 Table for more details.
(DOCX)

**S5 Table. Opinions on mask wearing by political affiliation.** * 0.10 ** 0.05 *** 0.01. OLS regressions with controls for own mask usage, gender, age, ethnicity, education, household income, the exchange rate, and the order of the PD games. Baseline group is Democrats. See S2 and S3 Tables for more details.
(DOCX)

**S6 Table. Relative moral opinions by political affiliation.** * 0.10 ** 0.05 *** 0.01. OLS regressions with controls for own mask usage, gender, age, ethnicity, education, household income, the exchange rate, and the order of the PD games. Baseline group is Democrats. See S4 Table for more details.
(DOCX)

**S7 Table. Marginal effects of political variables.** * 0.10 ** 0.05 *** 0.01. Errors clustered at individual level. Marginal effects from a Pooled Probit Regression using data on cooperation towards mask wearers, non-mask wearers and anonymous partners. Regressions in Panel A control for the interaction between political party and mask wearing partner; marginal effects are relative to that of being a Democrat. Regressions in Panel B control for the interaction between conservativeness and mask wearing partner. Also includes controls for gender, age, ethnicity, the political party supported, education, household income, the exchange rate, and the order of the PD games.
(DOCX)

**S8 Table. Interaction between political affiliation and partner type on mediators of cooperation.** * 0.10 ** 0.05 *** 0.01. Errors clustered at individual level. OLS regressions using data on expectations and altruism towards mask wearers and non-mask wearers. Includes controls for gender, age, ethnicity, education, household income, the session, and the order of the PD games.
(DOCX)

**S9 Table. Interaction between political conservativeness and partner type on mediators of cooperation.** * 0.10 ** 0.05 *** 0.01. Errors clustered at individual level. OLS regressions using data on expectations and altruism towards mask wearers and non-mask wearers. Includes controls for gender, age, ethnicity, education, household income, the session, and the order of the PD games.
(DOCX)

**S10 Table. Marginal effects within partner mask wearing conditions.** * 0.10 ** 0.05 *** 0.01. Errors clustered at individual level. Marginal effects from a Pooled Probit Regression using data on cooperation towards mask wearers, non-mask wearers and anonymous partners.

Regressions control for the interaction between political party and mask wearing partner, *and the interaction between own mask usage and mask wearing partner.* Marginal effects are relative to that of being mask wearer. Also, includes controls for gender, age, ethnicity, the political party supported, education, household income, the session, and the order of the PD games. (DOCX)

## Author Contributions

**Conceptualization:** Nattavudh Powdthavee, Yohanes E. Riyanto, Erwin C. L. Wong, Jonathan X. W. Yeo, Qi Yu Chan.

**Data curation:** Nattavudh Powdthavee, Yohanes E. Riyanto, Erwin C. L. Wong, Jonathan X. W. Yeo, Qi Yu Chan.

**Formal analysis:** Nattavudh Powdthavee, Yohanes E. Riyanto, Erwin C. L. Wong, Jonathan X. W. Yeo, Qi Yu Chan.

**Investigation:** Nattavudh Powdthavee, Yohanes E. Riyanto, Erwin C. L. Wong, Jonathan X. W. Yeo, Qi Yu Chan.

**Methodology:** Yohanes E. Riyanto, Erwin C. L. Wong, Jonathan X. W. Yeo, Qi Yu Chan.

**Writing – original draft:** Nattavudh Powdthavee, Yohanes E. Riyanto, Erwin C. L. Wong, Jonathan X. W. Yeo.

**Writing – review & editing:** Nattavudh Powdthavee, Yohanes E. Riyanto, Erwin C. L. Wong, Jonathan X. W. Yeo, Qi Yu Chan.

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
