## [Decision Letter · Decision Letter 0]

12 May 2021

PONE-D-21-13828

When Face Masks Signal Social Identity: Explaining the Deep Face-Mask Divide During the COVID-19 Pandemic

PLOS ONE

Dear Dr. Powdthavee,

Thank you for submitting your manuscript to PLOS ONE. After careful consideration, we feel that it has merit but does not fully meet PLOS ONE’s publication criteria as it currently stands. Therefore, we invite you to submit a revised version of the manuscript that addresses the points raised during the review process.

We look forward to receiving your revised manuscript.

Kind regards,

Jun Tanimoto

Academic Editor

PLOS ONE

Journal Requirements:

2. Please improve statistical reporting and refer to p-values as "p<.001" instead of "p=.000". Our statistical reporting guidelines are available at https://journals.plos.org/plosone/s/submission-guidelines#loc-statistical-reporting

[The project was funded by Warwick Business School and Nanyang Technological University personal research budgets.]

 [The author(s) received no specific funding for this work.]

4. Please include a copy of Table 2 which you refer to in your text on page 9.

Reviewers' comments:

Reviewer's Responses to Questions

**Comments to the Author**

1. Is the manuscript technically sound, and do the data support the conclusions?

Reviewer #1: Yes

Reviewer #2: Yes

2. Has the statistical analysis been performed appropriately and rigorously? 

Reviewer #1: Yes

Reviewer #2: Yes

3. Have the authors made all data underlying the findings in their manuscript fully available?

Reviewer #1: Yes

Reviewer #2: Yes

4. Is the manuscript presented in an intelligible fashion and written in standard English?

Reviewer #1: Yes

Reviewer #2: Yes

5. Review Comments to the Author

Reviewer #1: This MS reports a quite interesting lab-experimental to elucidate the relation between the social propensity of wearing-mask (not wearing-mask) and cooperative intention (defective intention) measure by Prisoner’s Dilemma (PD) game.

Unlike the previous studies which suggest that an individual who is (is not) compliant to wearing-mask could be regarded as cooperative (defective), what this work has successfully highlighted seems bit different. In a word, the reality laying behind is not a story that one can connect the attitude of wearing-mask (not-wearing-mask) with pro-social (anti-social), but more complex.

The point I can admire the authors’ approach is that they relied on a PD game to measure people’s potential extent of pro-sociality (or anti-sociality), which can be said as quite intrigue.

The authors found non-mask wearers were 2 less likely to cooperate than mask wearers when facing a mask-wearing partner, and more likely to cooperate than mask wearers when facing a non-mask wearing partner. As they insisting, these findings are surprising, considering that there is little evidence that mask wearers were generally more cooperative than non-mask wearers in the scenario where the information about mask usage was not known to the participants. In a sense, their experimental result suggest that, irrespective mask-wearer or non-mask-wearer, amid in-group, they have some sort of companion consciousness, which implies that a mask-wearer who has been regarded as cooperative tends to be defective to non-mask-wearer, while a non-mask-wearer who has been categorized as defective rather tends to cooperative to non-mask-wearer. Although there has been none of any hard evidence ever, such in-group attitude might be conceivable.

One thing of what their experiment found might be likely as below. The fact that a mask-wearer tends to be less cooperative to non-mask-wearer relates to the so-called ‘pro-social punishment’ scheme that has been widely studied in previous works of PD games. Likewise, the fact they elucidated, which a non-mask-wearer tends to be rather cooperative to non-mask-wearer than his attitude to mask-wearer, could be likened to ‘anti-social punishment’.

I would like to suggest the authors to add what I mentioned as above to their discussion so as to make it more affluent.

Another suggestion is that the authors should mention about rich stock of recent studies about social dilemma games concerning epidemic such as so-called ‘vaccination game’, especially OCVID-19. In Introduction part, the authors should review and cite several studies such as; (i) Sociophysics Approach to Epidemics, Springer, Springer, 2021, (ii) Social efficiency deficit deciphers social dilemmas, Scientific Reports 10, 16092, 2020, (iii) Evolutionary game theory modelling to represent the behavioural dynamics of economic shutdowns and shield immunity in the COVID-19 pandemic, Royal Society Open Science 7, 201095, 2020, and (iv) Vaccinating behaviour guided by imitation and aspiration, Proceedings of the Royal Society A 476, 2020327, 2020.

Reviewer #2: The authors attempt to explore a “Face Masks Signal Social Identity” works by doing some subjective experiment in which PD types are considered. The approach of this paper is interesting and can be applied in different locations to obtain important information about the self-protection measures (mask use, social distancing, hand washing, etc.). However, I have quite a few queries regarding the approach of this study and the way the study is carried out.

#1

The questionnaire, and abstract of this manuscript include PD game. However, in the whole manuscript we haven't found any quantitative results related to the PD game. A proper explanation in this respect is needed.

#2

Introduction is too large. Remove unnecessary information from the introduction. Is it really necessary to explain test hypothesis in the introduction section (page 5 and 6)?

#3

If possible, please add one figure that can represent overall picture of study settings and findings for general readers.

#4

Why very little number of people were non-mask-wearer? It’s very difficult to understand the real scenario based on less amount of respondence. Author should clearly explain this point. Therefore, in the scientific point of view, this survey seems to me very narrow.

#5

Also, authors should explain details about limitation of their survey and analysis.

#6

Figure captions need to be improved (both main text and SI).

#7

The manuscript contains a number of confusing sentences that should be corrected because somewhat misleading to my mind:

Abstract-

“Policy makers should therefore take social perception of face masks into account when designing not only what public messages to deliver, but also who to deliver these messages.”

Result

“Yet, there is little evidence in the first stage of our experiment, i.e., before we elicited their face mask usage, that mask wearers were more prosocial than non-mask wearers in a series of simultaneous and sequential moves PD games with anonymous partners.”

“More specifically, we found mask wearers were not only more altruistic towards other mask wearers, but they were also more likely to express a belief that other mask wearers are more likely to cooperate with them and expect cooperation from them than non-mask wearers in the PD game; see Figure A.1 in the SI.”

#8

Finally, Authors should read and cite the following literatures.

Influence of bolstering network reciprocity in the evolutionary spatial Prisoner’s Dilemma game: a perspective, Eur. Phys. J. B., 91: 312, 2018.

Hypothetical assessment of efficiency, willingness-to-accept and willingness-to-pay for dengue vaccine and treatment: a contingent valuation survey in Bangladesh, Human vaccine and Immunotherapeutics, DOI: 10.1080/21645515.2020.1796424 (2020).

“Do humans play according to the game theory when facing the social dilemma situation?” A survey study, EVERGREEN, 07(01), 7-14 (2020).

6. PLOS authors have the option to publish the peer review history of their article (what does this mean?). If published, this will include your full peer review and any attached files.

Reviewer #1: No

Reviewer #2: No

---

## [Author Response · Author response to Decision Letter 0]

23 May 2021

Reviewer #1: This MS reports a quite interesting lab-experimental to elucidate the relation between the social propensity of wearing-mask (not wearing-mask) and cooperative intention (defective intention) measure by Prisoner’s Dilemma (PD) game.

Unlike the previous studies which suggest that an individual who is (is not) compliant to wearing-mask could be regarded as cooperative (defective), what this work has successfully highlighted seems bit different. In a word, the reality laying behind is not a story that one can connect the attitude of wearing-mask (not-wearing-mask) with pro-social (anti-social), but more complex.

The point I can admire the authors’ approach is that they relied on a PD game to measure people’s potential extent of pro-sociality (or anti-sociality), which can be said as quite intrigue.

The authors found non-mask wearers were 2 less likely to cooperate than mask wearers when facing a mask-wearing partner, and more likely to cooperate than mask wearers when facing a non-mask wearing partner. As they insisting, these findings are surprising, considering that there is little evidence that mask wearers were generally more cooperative than non-mask wearers in the scenario where the information about mask usage was not known to the participants. In a sense, their experimental result suggest that, irrespective mask-wearer or non-mask-wearer, amid in-group, they have some sort of companion consciousness, which implies that a mask-wearer who has been regarded as cooperative tends to be defective to non-mask-wearer, while a non-mask-wearer who has been categorized as defective rather tends to cooperative to non-mask-wearer. Although there has been none of any hard evidence ever, such in-group attitude might be conceivable.

One thing of what their experiment found might be likely as below. The fact that a mask-wearer tends to be less cooperative to non-mask-wearer relates to the so-called ‘pro-social punishment’ scheme that has been widely studied in previous works of PD games. Likewise, the fact they elucidated, which a non-mask-wearer tends to be rather cooperative to non-mask-wearer than his attitude to mask-wearer, could be likened to ‘anti-social punishment’.

I would like to suggest the authors to add what I mentioned as above to their discussion so as to make it more affluent.

**Thank you very much for taking the time to read our paper. We are very glad to know that you like our paper and have now incorporated the above suggestions in our discussion of the residual effect on p.14.

Another suggestion is that the authors should mention about rich stock of recent studies about social dilemma games concerning epidemic such as so-called ‘vaccination game’, especially OCVID-19. In Introduction part, the authors should review and cite several studies such as; (i) Sociophysics Approach to Epidemics, Springer, Springer, 2021, (ii) Social efficiency deficit deciphers social dilemmas, Scientific Reports 10, 16092, 2020, (iii) Evolutionary game theory modelling to represent the behavioural dynamics of economic shutdowns and shield immunity in the COVID-19 pandemic, Royal Society Open Science 7, 201095, 2020, and (iv) Vaccinating behaviour guided by imitation and aspiration, Proceedings of the Royal Society A 476, 2020327, 2020.

**Thank you very much for your suggestion about the vaccination game. We have now included a few of the recommended citations that we believe to be relevant to this study at the end of the introduction.

———

Reviewer #2: The authors attempt to explore a “Face Masks Signal Social Identity” works by doing some subjective experiment in which PD types are considered. The approach of this paper is interesting and can be applied in different locations to obtain important information about the self-protection measures (mask use, social distancing, hand washing, etc.). However, I have quite a few queries regarding the approach of this study and the way the study is carried out.

#1

The questionnaire, and abstract of this manuscript include PD game. However, in the whole manuscript we haven't found any quantitative results related to the PD game. A proper explanation in this respect is needed.

**Thank you very much for taking the time to carefully read our paper. We have now made the reference to the PD game more explicit by incorporating a section on the experimental methodology. The section explains how we use the PD game to measure cooperation by mask wearers and non-mask wearers which is described in the results section.

#2

Introduction is too large. Remove unnecessary information from the introduction. Is it really necessary to explain test hypothesis in the introduction section (page 5 and 6)?

**Thank you for this point. We have now moved the hypotheses section to the Materials and Methods section.

#3

If possible, please add one figure that can represent overall picture of study settings and findings for general readers.

**Done.

#4

Why very little number of people were non-mask-wearer? It’s very difficult to understand the real scenario based on less amount of respondents. Author should clearly explain this point. Therefore, in the scientific point of view, this survey seems to me very narrow.

**The main explanation for the relatively small number of non-mask wearers in our sample (N=59) is because we ran our experiment in December 2020 and, by that time, there was already around 70-80% of face mask take-up rate in America. Hence, our relatively smaller number reflects fairly representatively the true number of non-mask wearers at the time of carrying out the survey. Had we ran our experiment in early 2020, we would have been able to recruit a significantly higher number of non-mask wearers for our analysis. We have now discussed this point explicitly on p.5.

#5

Also, authors should explain details about limitations of their survey and analysis.

**Done. Please find this in the conclusion section.

#6

Figure captions need to be improved (both main text and SI).

**We would be happy to do so. However, we are not sure how the figure captions can be further improved. Currently, we believe that our figure and table captions are concise and informative. If you could be more specific about the changes you want us to do, we would be more than happy to oblige. 

#7

The manuscript contains a number of confusing sentences that should be corrected because somewhat misleading to my mind:

Abstract-

“Policy makers should therefore take social perception of face masks into account when designing not only what public messages to deliver, but also who to deliver these messages.”

Result

“Yet, there is little evidence in the first stage of our experiment, i.e., before we elicited their face mask usage, that mask wearers were more prosocial than non-mask wearers in a series of simultaneous and sequential moves PD games with anonymous partners.”

“More specifically, we found mask wearers were not only more altruistic towards other mask wearers, but they were also more likely to express a belief that other mask wearers are more likely to cooperate with them and expect cooperation from them than non-mask wearers in the PD game; see Figure A.1 in the SI.”

**We have since rewritten these sentences to make them clearer to understand.

#8

Finally, Authors should read and cite the following literature.

Influence of bolstering network reciprocity in the evolutionary spatial Prisoner’s Dilemma game: a perspective, Eur. Phys. J. B., 91: 312, 2018.

Hypothetical assessment of efficiency, willingness-to-accept and willingness-to-pay for dengue vaccine and treatment: a contingent valuation survey in Bangladesh, Human vaccine and Immunotherapeutics, DOI: 10.1080/21645515.2020.1796424 (2020).

“Do humans play according to the game theory when facing the social dilemma situation?” A survey study, EVERGREEN, 07(01), 7-14 (2020).

**Thank you very much for your suggestions. We have read the suggested papers, but could only find one - the EVERGREEN publication - to be relevant to our study. We’ve decided to cite only the EVERGREEN publication, but please do not hesitate to let us know why you think we should be citing the other two recommended studies here as well.

---

## [Decision Letter · Decision Letter 1]

31 May 2021

When Face Masks Signal Social Identity: Explaining the Deep Face-Mask Divide During the COVID-19 Pandemic

PONE-D-21-13828R1

Dear Dr. Powdthavee,

We’re pleased to inform you that your manuscript has been judged scientifically suitable for publication and will be formally accepted for publication once it meets all outstanding technical requirements.

Kind regards,

Jun Tanimoto

Academic Editor

PLOS ONE

Additional Editor Comments (optional):

Reviewers' comments:

Reviewer's Responses to Questions

**Comments to the Author**

1. If the authors have adequately addressed your comments raised in a previous round of review and you feel that this manuscript is now acceptable for publication, you may indicate that here to bypass the “Comments to the Author” section, enter your conflict of interest statement in the “Confidential to Editor” section, and submit your "Accept" recommendation.

Reviewer #1: All comments have been addressed

Reviewer #2: (No Response)

2. Is the manuscript technically sound, and do the data support the conclusions?

Reviewer #1: Yes

Reviewer #2: Yes

3. Has the statistical analysis been performed appropriately and rigorously? 

Reviewer #1: Yes

Reviewer #2: Yes

4. Have the authors made all data underlying the findings in their manuscript fully available?

Reviewer #1: Yes

Reviewer #2: Yes

5. Is the manuscript presented in an intelligible fashion and written in standard English?

Reviewer #1: Yes

Reviewer #2: Yes

6. Review Comments to the Author

Reviewer #1: The authors seemed tackling to respond all suggestions I gave, thus, now the revised MS seems to be acceptable.

Reviewer #2: The authors improved the manuscript taking into account most of the comments and advice I wrote in my first report. Therefore, I am happy to recommend now the paper in the current form.

7. PLOS authors have the option to publish the peer review history of their article (what does this mean?). If published, this will include your full peer review and any attached files.

Reviewer #1: No

Reviewer #2: No

---

## [Editor Report · Acceptance letter]

3 Jun 2021

PONE-D-21-13828R1 

When face masks signal social identity: Explaining the deep face-mask divide during the COVID-19 pandemic 

Dear Dr. Powdthavee:

I'm pleased to inform you that your manuscript has been deemed suitable for publication in PLOS ONE. Congratulations! Your manuscript is now with our production department. 

Kind regards, 

on behalf of

Prof. Jun Tanimoto 

Academic Editor

PLOS ONE